# PRIORITIZED TRAINING ON POINTS THAT ARE LEARNABLE, WORTH LEARNING, AND NOT YET LEARNED

## ABSTRACT

We introduce *reducible held-out loss selection* (RHOLS), a technique for faster model training which selects training points that are "just right". We propose an information-theoretic selection function and a tractable, easy-to-implement approximation—the reducible held-out loss—to efficiently choose training points that maximize information about a holdout set. We show that "hard" (e.g. high loss) points usually selected in the optimization literature are typically noisy, leading to deterioration on real-world, noisy datasets. At the same time, "easy" (e.g. low noise) samples often prioritized for curriculum learning confer less information. In contrast, RHOLS chooses points that are "just right" and trains in fewer steps than the above approaches.

## 1 INTRODUCTION

State-of-the-art models such as GPT-3 (Brown et al., 2020), CLIP (Radford et al., 2021), and ViT (Dosovitskiy et al., 2021) achieve remarkable results by training on vast amounts of web scraped data. Especially for large models, training on such datasets requires excessive training times that are measured in weeks or months rather than hours (Radford et al., 2021; He et al., 2020). Even practitioners who work with modestly sized models face long development cycles, due to numerous iterations of algorithm design and hyperparameter selection. These considerations highlight that the total time required for training is a core constraint in the development of such deep learning models.

To speed up training, practitioners with sufficient resources use many more machines and much larger batches. However, there are rapidly diminishing returns to using larger batches and distributing them across more machines (LeCun et al., 2012; Keskar et al., 2016), to a point where adding machines does not reduce training time (Anil et al., 2018).

At the same time, practitioners realized that not all real-world data samples are equally useful. Many samples are *noisy*, i.e. mislabelled or inherently ambiguous. For example, the text associated with a web scraped image is rarely an accurate description of the image. Other samples are *redundant*, for example due to the typical overrepresentation of certain object classes in web scraped data (Tian et al., 2021). Such redundant samples can often be left out without losing performance. In many cases, data is so abundant that state-of-the-art performance is reached in less than a single epoch (Brown et al., 2020; Kaplan et al., 2020). With such abundance, it can easily be afforded to skip less useful points.

There are different approaches to selecting the most useful datapoints for training. Some bodies of literature, including curriculum learning, suggest prioritizing *easy* points with low label noise before training on all points equally (Bengio et al., 2009). While this can improve convergence and generalization, it has no mechanism to avoid points that are already learned (redundant). Other works instead suggest to train on points that are *hard* for the current model. In the optimization literature, online batch selection methods (Loshchilov & Hutter, 2015) do this by selecting points with high loss or high gradient norm. Such online selection can successfully skip points that are already learned and focus on the few points where it is possible to make the greatest learning progress (Figure 1, left).

However, we show that prioritising hard examples can be detrimental on noisy data. In real-world datasets, high loss examples may be mislabelled or inherently ambiguous. Indeed, in controlled experiments, with just 10% uniform label noise, points selected by these methods are overwhelmingly those where the label is corrupted (Figure 1, right). Our results show that prioritising them degrades performance severely. Additionally, some samples are hard because they are outliers with unusual

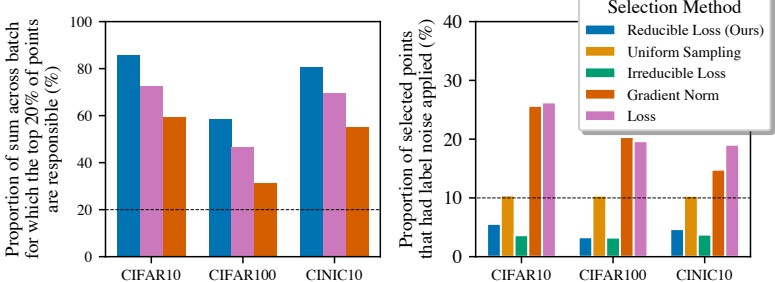

Figure 1: Properties of different training data selection functions: reducible loss (ours), irreducible loss, loss, and gradient norm. **Left:** We score the points in each batch with each selection function and sum the score of the top 20% of points, then divide it by the total sum across all points. Across selection functions, the set of top-scoring points is responsible for most of the total; this suggests that many points are already learned (redundant) and can be skipped. **Right:** Proportion of selected points with label noise applied. We added 10% uniform label noise, i.e., we randomly switched each point's label with 10% probability. Selecting the 10% "hardest" points (high loss or gradient norm) gives batches where > 10% of the labels are mislabeled. We can avoid noisy points by selecting points that are "easy" (low irreducible loss) or have high reducible loss. Results are averaged over 15 epochs.

features, far from the other training samples. This type of difficult point can also lie far from the test data, which makes it less *relevant* for reducing the test loss.

To overcome these limitations, we introduce *reducible held-out loss selection* (RHOLS). We introduce a selection function—Predictive Information Content (PIC)—that quantifies the information each point confers about the labels of a holdout set. Approximating this, we derive an intuitive, tractable, and easy-to-implement novel selection function: reducible held-out (cross-entropy) loss. By maximizing the information gained about the labels of a held-out set, our method chooses non-redundant, relevant points with low noise—points that are "just right".

We explore the effects of reducible loss selection on five datasets, with and without additional label noise. We evaluate the reduction in required training steps compared to conventional uniform sampling and recent selection methods. Our method reaches target accuracy in 1.76x-3.82x fewer steps than conventional uniform sampling on CIFAR-10, CIFAR-100, and CINIC-10. In the presence of just 10% label noise, this speedup increases further to 2.34x-5.15x while the speed of state-of-the art prior methods degrades severely. On Clothing-1M, a large-scale dataset with noisy, web-scraped labels, our method reaches target accuracy in 12.5x fewer steps compared to uniform sampling and a 8% higher final accuracy.

## 2 BACKGROUND: ONLINE BATCH SELECTION

Consider a model $p(y \mid x, \theta)$ trained on data $\mathcal{D} = \{(x_i, y_i)\}_{i=1}^{n}$ using stochastic gradient descent. At each training step $t$, we load a batch $b_t$ of size $n_b$ from $\mathcal{D}$. In online batch selection (Loshchilov & Hutter, 2015), we pre-sample a larger batch $B_t$ of size $n_B > n_b$. Then, we construct a smaller batch $b_t$ that consists of the top-ranking $n_b$ points in $B_t$ ranked by a label-aware selection function $S(x_i, y_i)$. We perform one gradient step to minimize a mini-batch loss $\sum_{i \in b_t} L(y_i, p(y_i \mid x_i, \theta))$. The following large batch $B_{t+1}$ is then pre-sampled from $\mathcal{D}$ without replacement of the previously sampled points (points are replaced only at the start of the next epoch).

We work with this approach for its simplicity, convergence guarantees, and strong performance in recent work (Kawaguchi & Lu, 2020). Further, it could be extended to incorporate stochastic data selection (Jiang et al., 2019) with importance weights (Schaul et al., 2015; Katharopoulos & Fleuret, 2018) or to include earlier points $x_i$ in $B_t$ whose selection score $S(x_i, y_i)$ has been previously computed (Loshchilov & Hutter, 2015). Without importance weights, the simple approach above biases the location of the minimum of the loss. However, this biased selection can improve test performance both in theory and practice (Farquhar et al., 2021; Kawaguchi & Lu, 2020).

**Notation.** For a holdout set $\{y_i^{\text{ho}}\}_i$, $\{x_i^{\text{ho}}\}_i$ we will write $\mathbf{y}^{\text{ho}}$, $\mathbf{x}^{\text{ho}}$ for brevity. Lower-case variables like $y$ are outcomes and upper-case variables like $Y$ are random variables.

## 3    REDUCIBLE HELD-OUT LOSS—A MEASURE OF PREDICTIVE INFORMATION CONTENT

Online batch selection methods construct batches at training time using points that have high loss or high gradient norm (Loshchilov & Hutter, 2015; Katharopoulos & Fleuret, 2018). In this section, we derive and discuss an alternative selection function, based on an information-theoretic perspective.

**Predictive Information Content (PIC).** Our overall objective is to select the point $(x, y) \in \mathcal{D}$ for training that minimizes the expected loss $\mathbb{E}_{\text{P}_{\text{true}}(x,y)} \text{h}[y \mid x, \mathcal{D}_\text{t} \cup (x, y)]$ of the model, where

$$\text{h}[y \mid x, \mathcal{D}_\text{t}] := -\log \text{p}(y \mid x, \mathcal{D}_\text{t}) \tag{1}$$

denotes the cross-entropy loss on $(x, y)$ of a model trained on $\mathcal{D}_\text{t}$. As a proxy, we aim to maximize the reduction in the Monte Carlo expectation of the cross-entropy loss on a held-out dataset $\mathcal{D}_{\text{ho}}$,

$$\frac{1}{|\mathcal{D}_{\text{ho}}|} \sum_{i \in \mathcal{D}_{\text{ho}}} \text{h}[y_i^{\text{ho}} \mid x_i^{\text{ho}}, \mathcal{D}_\text{t}] - \text{h}[y_i^{\text{ho}} \mid x_i^{\text{ho}}, \mathcal{D}_\text{t} \cup (x, y)]. \tag{2}$$

For a deterministic model (i.e. with fixed parameters), this held-out loss (to a constant factor) equals

$$\text{h}[\mathbf{y}^{\text{ho}} \mid \mathbf{x}^{\text{ho}}, \mathcal{D}_\text{t}] - \text{h}[\mathbf{y}^{\text{ho}} \mid \mathbf{x}^{\text{ho}}, \mathcal{D}_\text{t} \cup (x, y)]. \tag{PIC-1} \tag{3}$$

We term this the Predictive Information Content (PIC). It quantifies *predictive* information: the part of the information gained by observing $(x, y)$ that is applicable to predicting the holdout labels. It also relates to the Expected Predictive Information Gain (EPIG) recently proposed for active learning (Kirsch et al., 2021). In active learning, one selects points without access to their labels, and therefore Kirsch et al. (2021) must compute the expectation of (3) over a posterior predictive $\text{p}(\mathbf{Y}^{\text{ho}} \mid \mathbf{x}^{\text{ho}}, \mathcal{D}_t)$. However, here we quantify not the expected but the actual information gained when observing $(x, y)$. We therefore follow naming conventions and term (3) a (predictive) information content. See Appendix B for more details on the relationship to active learning and EPIG.

**Deriving a tractable selection function: the irreducible and reducible losses.** The two terms in (3) cannot be computed efficiently; they require first training the model on each potential point $(x, y)$ and then performing a forward pass with the model on the full holdout set. We derive an alternative, more tractable expression. To make our claims precise and our assumptions transparent, we use the language of Bayesian probability theory to formalise our claims. In the Bayesian context, we treat model parameters as a random variable with prior $\text{p}(\theta)$ and infer a posterior $\text{p}(\theta|\mathcal{D}_\text{t})$ using the sequence of already-seen training data $\mathcal{D}_\text{t} = b_{1:t-1}$. The model has a predictive distribution $\text{p}(y|x, \mathcal{D}_\text{t}) = \int_\theta \text{p}(y|x, \theta) \, \text{p}(\theta|\mathcal{D}_\text{t}) d\theta$. When using a point estimate of $\theta$ (such as an MLE or MAP), the predictive distribution can still be written as an integral, but with respect to a Dirac delta at the point estimate as an approximation of the true posterior. Applying Bayes rule to $\text{p}(\mathbf{y}^{\text{ho}} \mid y, \mathbf{x}^{\text{ho}}, x, \mathcal{D}_\text{t})$ and using the conditional independence assumption $\text{p}(y_i \mid x_i, x_j, \mathcal{D}_\text{t}) = \text{p}(y_i \mid x_i, \mathcal{D}_\text{t})$, we can reformulate PIC as

$$\text{h}[y \mid x, \mathcal{D}_\text{t}] - \text{h}[y \mid x, \mathbf{y}^{\text{ho}}, \mathbf{x}^{\text{ho}}, \mathcal{D}_\text{t}], \tag{PIC-2} \tag{4}$$

moving the holdout set to the conditional. We show this derivation in Appendix A.

The first term is simply the cross-entropy loss on the potential point $(x, y)$ of a model trained on $\mathcal{D}_\text{t}$. The second term—the cross-entropy loss of a model trained on $\mathcal{D}_\text{t}$ *and the held-out set*—adds computation as it requires constant updating with increasing $\mathcal{D}_\text{t}$. We can approximate this second term by training a model only on the held-out dataset, and refer to it as the *irreducible held-out loss*:

$$\text{h}[y \mid x, \mathbf{y}^{\text{ho}}, \mathbf{x}^{\text{ho}}]. \tag{irreducible held-out loss} \tag{5}$$

This approximation may be reasonable when $|\mathcal{D}_\text{t}| \ll |\mathcal{D}_{\text{ho}}|$ (e.g. at the start of training) or when a small model $\text{p}(y \mid \mathbf{y}^{\text{ho}}, \mathbf{x}^{\text{ho}})$ trained on $\mathcal{D}_{\text{ho}}$ has limited capacity to improve further by training on $\mathcal{D}_\text{t}$. Empirically, we find this approximation does not affect performance (Appendix D). We refer to

Eq. (5) as the irreducible held-out loss since it is the portion of the loss that could not be reduced by training on the held-out set. Plugging this approximation into Eq. (4), we get the *reducible held-out loss*,

$$\mathrm{h}[y \mid x, \mathcal{D}_{\mathrm{t}}] - \mathrm{h}[y \mid x, \mathbf{y}^{\mathrm{ho}}, \mathbf{x}^{\mathrm{ho}}]. \qquad \text{(reducible held-out loss)} \quad (6)$$

This provides an easy-to-compute selection function that approximates the Predictive Information Content between $\mathcal{D}_{\mathrm{ho}}$ and the datapoint $(x, y)$ for a model trained on $\mathcal{D}_{\mathrm{t}}$.

**Understanding reducible loss.** We now provide intuition on why the reducible held-out loss selects non-redundant, non-noisy points. The reducible held-out loss contains two terms. The first term, the loss, is high on points difficult for the current model i.e. points that are not yet learned and thus not redundant. Indeed, existing online batch selection methods use loss for selection (Loshchilov & Hutter, 2015). However, not all high loss points are informative—some may be noisy or mislabelled. Crucially, these points cannot be predicted using the holdout set and thus have high irreducible loss. Therefore, these non-information points have low reducible loss. Indeed, we show that the cross-entropy loss disproportionally selects such noisy points in Figure 1 whereas selecting points with high reducible loss (or low irreducible loss) avoids noisy points. In short, RHOLS prioritises points that are not redundant (high loss) and not noisy (low irreducible loss).

Loss based selection has additional pitfalls. The loss is likely higher for outliers—values of $x$ far from the training data, such as those in a region with low input density $\mathrm{p}_{\mathrm{true}}(x)$. Since our aim is to reduce the expected loss $\mathbb{E}_{\mathrm{p}_{\mathrm{true}}(x,y)}[\mathrm{h}[y \mid x, \mathcal{D}_{\mathrm{t}}]]$ under $\mathrm{p}_{\mathrm{true}}(x, y)$, we do not want to prioritize less relevant outliers. Since such points are also less likely in the held-out data, the irreducible loss can also be expected to be higher. Thus, reducible held-out loss should select points that are more 'relevant' to the holdout set. We find empirical evidence for this below.

**Selecting whole batches.** We developed the theory assuming the selection of a single point $(x, y)$. When selecting entire batches, we select the points with the top-$n_b$ scores from a randomly sampled set $B_t$ of candidate examples. This assumes that each point has little effect on the scores of other points, and is often used as a simplifying assumption for selection in the context of active learning (Kirsch et al., 2019). This is a much more reasonable assumption in our case than in active learning, because model predictions are not changed much by a single gradient step on one mini-batch.

Algorithm 1 shows the implementation of batch selection using the reducible held-out loss. Note we precompute and store the irreducible losses before training. This way, we avoid recomputing them in every epoch and across different training runs.

---

**Algorithm 1** Reducible held-out loss selection (RHOLS)

---

1: **Input:** Learning rate $\eta$, model $\mathrm{p}(y \mid x, \mathcal{D}_{\mathrm{ho}})$ trained on a holdout set $\mathcal{D}_{\mathrm{ho}}$, batch size $n_b$, large batch size $n_B > n_b$.

2: **for** $(x_i, y_i)$ in `training set` **do**
3:     `IrreducibleLoss[i]` $\leftarrow \mathrm{h}[y_i \mid x_i, \mathcal{D}_{\mathrm{ho}}])$
4: **end for**

    Initialize parameters $\theta^0$ and $t = 0$
5: **for** $t = 1, 2, \ldots$ **do**
6:     Randomly select a large batch $B_t$ of size $n_B$.
7:     $\forall i \in B_t$, compute `Loss[i]`, the loss of point $i$ given parameters $\theta^t$
8:     $\forall i \in B_t$, compute `ReducibleLoss[i]` $\leftarrow$ `Loss[i]` $-$ `IrreducibleLoss[i]`
9:     $b_t \leftarrow$ top-$n_b$ samples in $B_t$ in terms of `ReducibleLoss`.
10:     $g_t \leftarrow$ mini-batch gradient on $b_t$ using parameters $\theta^t$
11:     $\theta^{t+1} \leftarrow \theta^t - \eta g_t$

---

**Simple parallelized selection.** For large-scale neural network training, practitioners with sufficient resources would use many more machines if it further sped up training (Anil et al., 2018). However, as more workers are added in synchronous or asynchronous gradient descent, the returns diminish to a point where adding more workers does not further improve wall clock time (Anil et al., 2018). For

example, there are rapidly diminishing returns for using larger batch sizes or distributing a given batch across more workers (LeCun et al., 2012; Keskar et al., 2016). The same holds for distributing the model across more workers along its width or depth (Rasley et al., 2020; Shoeybi et al., 2019; Huang et al., 2019). However, we can circumvent these diminishing returns by adding a new dimension of parallelization, namely, for data selection.

Since parallel *forward* passes do not suffer from such diminishing returns, one can easily use extra workers for them. The theoretical runtime speedup can be calculated as follows. The computational cost of selection is $\frac{n_b}{3n_B}$ times as much as the cost of a training step on $b_t$. This is because forward passes require 3x less computation than a full forward-backward pass (Jouppi et al., 2017). One can accelerate this computation almost arbitrarily by adding more workers that compute losses using a copy of the model being trained. The limit is reached when the time for selection is dominated by the cost of communicating parameter updates to workers. More sophisticated parallelization strategies allow reducing the time overhead even further, see Section 4. To avoid assumptions about the particular strategy used, we report experiment results in terms of the required number of steps.

# 4  RELATED WORK

**Data selection functions.**    Our reducible loss is best understood as an alternative to other selection functions. Such metrics can be categorized by the properties of points they select and by whether they use information about labels. Both high loss (Loshchilov & Hutter, 2015; Kawaguchi & Lu, 2020; Jiang et al., 2019) and high prediction uncertainty (Settles, 2009; Li & Sethi, 2006; Gal et al., 2017; Coleman et al., 2020) select for points that are "difficult" for the current model. But while loss is only applicable to labelled data, prediction uncertainty is designed for the active learning setting where labels are not available. Despite this difference, due to their similarity they face the same problem: high loss and high uncertainty can be due to an ambiguous ground-truth or a noisy label. A related technique prioritizes samples whose labels are easily forgotten during training (Toneva et al., 2018).

**Variance reduction methods.**    Online sample selection methods have also been suggested as a way to reduce the variance of the gradient estimator computed on mini-batches (Katharopoulos & Fleuret, 2018; 2017; Johnson & Guestrin, 2018; Alain et al., 2015). Such methods typically use importance sampling, where points with high gradient norm (or approximations thereof) are sampled with high probability. To avoid biasing the gradient estimator, the gradient of high-probability points is then down-weighted. Importance sampling can be effective but also sensitive to hyperparameters (Johnson & Guestrin, 2018). Furthermore, Figure 1 (right) shows that the gradient norm is also often highest for noisy points, which may make it unsuitable for noisy real-world datasets.

**Computationally efficient data selection.**    While we limit our analysis to the choice of selection function and compute the metrics naively, this choice is likely inefficient in practice. Selection can be made cheaper by reusing losses computed in previous epochs (Loshchilov & Hutter, 2015; Jiang et al., 2019) or training a small model to predict them (Katharopoulos & Fleuret, 2017; Zhang et al., 2019). Alternatively, core set methods only perform selection once before training (Mirzasoleiman et al., 2020; Borsos et al., 2020). Such methods typically use expensive metrics that are not economical for large web scraped datasets where training typically only lasts a few epochs.

**Time-efficient data selection.**    To accelerate data selection, prior work uses a set of workers that perform forward passes on $B_t$ or on the entire dataset asynchronously while the master process trains on recently selected data (Alain et al., 2015; Schaul et al., 2015). This supplements rather than replaces standard data parallelism, a method whose returns diminish as more workers are added (Anil et al., 2018). Furthermore, forward passes can be accelerated using low-precision hardware. A forward pass by default already requires roughly 3x less time than a forward-backward pass but this speedup can be increased to a factor around 10 when using the low precision cores available in modern GPUs and TPUs (Jouppi et al., 2017; Jiang et al., 2019). While backward passes typically require higher precision, forward passes can tolerate lower precision.

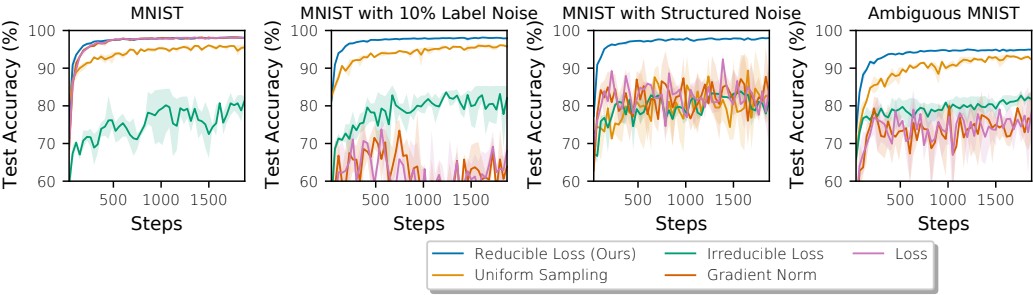

Figure 2: Reducible loss is robust to a variety of label noise patterns, while other selection methods degrade. A step corresponds to lines $6 - 11$ in Algorithm 1. Lines correspond to means and shaded areas to minima and maxima across 3 random seeds.

## 5 EXPERIMENTS

**Datasets.** We evaluate on five datasets. 1) QMNIST (Yadav & Bottou, 2019) augments the MNIST dataset (LeCun et al., 1998) with an additional 50,000 images. We make use of the additional 50,000 images as the holdout set. 2) On CIFAR-10 (Krizhevsky & Hinton, 2009) we train on half of the training set and use the other half as a holdout set used for training the irreducible loss model. 3) CIFAR-100: same as CIFAR-10. 4) CINIC-10 (Darlow et al., 2018) has 4.5x more images than CIFAR-100 and includes a holdout set and a test set with 90,000 images each. 5) Clothing-1M (Xiao et al., 2015), which contains over 1 million clothing images divided into 14 classes. The data is web-scraped—a key application area of our work–and thus has noisy labels. The dataset also contains 50k, 14k, and 10k images with clean labels for training, validation, and testing, respectively. We use the 50k clean training samples to train our irreducible loss model, and include the 50k clean training samples in the larger dataset when running the baselines. As a web-scraped dataset, Clothing-1M is most representative of the setting where training time rather than data is the bottleneck.

**Baselines.** The most important baseline is the *de facto* standard in deep learning, uniform selection (without replacement). We also evaluate against selection functions that have achieved competitive performance recently: the plain cross-entropy loss (as implemented by Kawaguchi & Lu (2020)), gradient norm, and gradient norm with importance sampling (called *gradient norm IS* in our figures) (Katharopoulos & Fleuret, 2018). We follow prior work by approximating the per-example gradient with the last layer gradient norm for computational tractability. We also compare to the core-set method Selection-via-Proxy (SVP) (Coleman et al., 2020) that selects data offline before training to probe if maximal performance requires online selection. We report results using maximum entropy SVP and select with the best-performing model. We compare only to alternative selection functions. Other improvements such as predicting or reusing losses (Katharopoulos & Fleuret, 2017; Zhang et al., 2019; Jiang et al., 2019) are orthogonal to our contribution and may be combined with our selection function to further accelerate training.

We use a 3 layer MLP for experiments on QMNIST, and a ResNet-18 for all other datasets. We use the Pytorch default hyperparameters and the AdamW optimiser. Please see Appendix C for details.

### 5.1 ROBUSTNESS TO NOISE

We begin by evaluating the performance of different selection methods under a variety of noise patterns on QMNIST and variations thereof. We use this dataset because it has little label noise in its original form, allowing us to test the effect of adding noise. Firstly, we add uniform label noise to 10% of training points. Secondly, we add structured label noise that affects easily confused classes. We follow Rolnick et al. (2017) and flip the labels of the four most frequently confused classes (in the confusion matrix of a trained model) with 50% probability. For example, a 2 is often confused with a 5; thus we change the label of all 2s to 5s with 50% probability. Thirdly, we leverage the natural noise distribution of MNIST by using AmbiguousMNIST (Mukhoti et al., 2021) as the training set. AmbiguousMNIST contains a training set with 60k generated ambiguous digits that have more than one plausible label.

While selecting with loss and gradient norm trains accelerates training on the MNIST training set, their performance degrades on all three types of noise distributions (Figure 2). This is in line with our finding in Figure 1 (right) that they over-select points with noisy labels. In contrast, reducible held-out loss is robust to label noise and trains in fewer steps than all baselines.

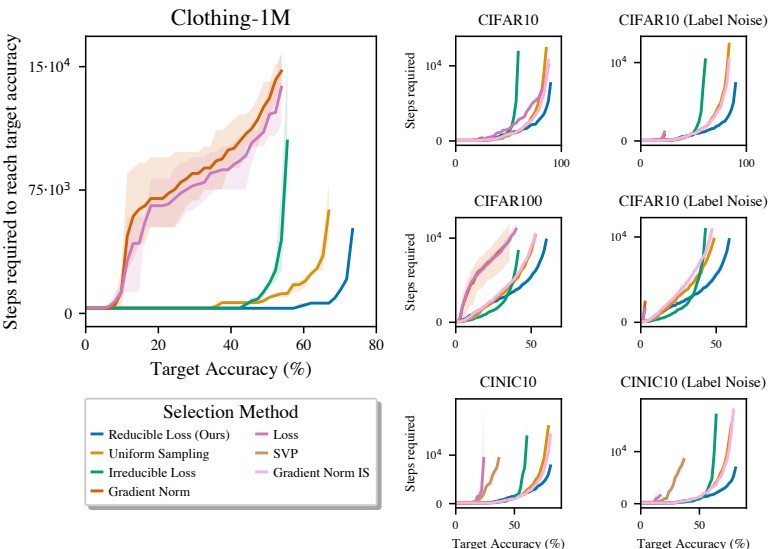

Figure 3: Gradient steps required to achieve a given test accuracy (lower is better). **Left column:** The speedup of RHOLS over uniform sampling is greatest on a large-scale web scraped dataset with noisy labels (12.5x). **Middle column:** Speedups are still substantial on clean datasets. **Right column:** Applying 10% uniform label noise to training data degrades other methods but increases the speedup of our method. A step corresponds to lines 6 − 11 in Algorithm 1. Lines correspond to means and shaded areas to minima and maxima across 3 random seeds.

## 5.2 SPEEDUP IN STEPS REQUIRED

Here, we evaluate how many steps each selection method needs to achieve a given test accuracy.

As our focus is on evaluating a new selection function rather than an entire training pipeline, we report the number of steps needed to reach a target accuracy. How the required steps relate to the wall clock time depends on implementation details that are beyond the scope of this paper. Most importantly, data selection is amenable to parallelization beyond standard data parallelism as discussed in Sections 3 and 4. Furthermore, a model trained on a holdout set can be reused for selection during the many runs that are part of a training pipeline.

**Speedup on clean data.** RHOLS reaches the same accuracies in fewer steps than standard uniform selection without replacement across all datasets (Figure 3, top). Furthermore, it outperforms all baselines across all datasets. The observed speedup with respect to uniform selection is 3.82x on CIFAR-10, 1.76x on CIFAR-100, and 2.88x on CINIC-10.

**Speedup on noisy data.** We uniformly perturb the labels for 10% of the datapoints. Under label noise, batch selection with reducible held-out loss further increases its speedup over the other methods (Figure 3, bottom). Particularly interesting is that, on noisier data, the speedup over uniform selection also grows. The speedups we observe relative to uniform selection are 5.15x on CIFAR-10, 2.34x on CIFAR-100, and 4.17x on CINIC-10. This aligns with our information-theoretical perspective on reducible held-out loss and with the finding in Figure 1 (right) that reducible held-out loss avoids noisy points while other methods prefer them.

**Speedup on large-scale web-scraped data.** On Clothing-1M, no baseline matches uniform selection, suggesting they are not robust to noise. In contrast, RHOLS reaches the accuracy that uniform

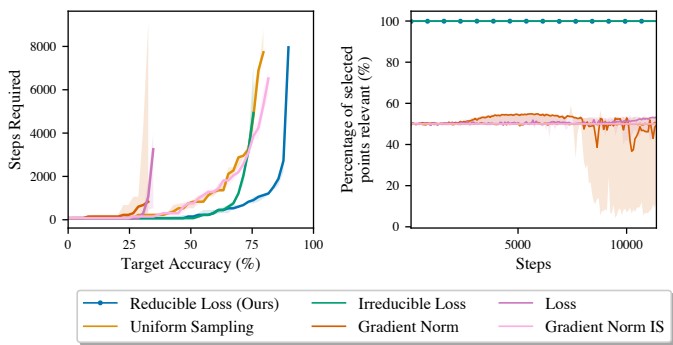

Figure 4: CINIC-10 with half of the classes absent (irrelevant) in the test and holdout sets. **Left:** steps required to reach a given target accuracy on the test set. **Right:** Proportion of selected points whose labels do appear in the holdout and test sets. RHOLS and irreducible loss select exclusively the relevant classes. ROHLS provides a large speedup over existing baselines. Lines correspond to means and shaded areas to standard deviations across 3 random seeds

selection achieves after epoch 50 in just 2 epochs, providing a speedup of 12.5x. Furthermore, RHOLS' reaches a higher final accuracy (74%) than uniform selection (66%).

**Reducible held-out loss can select more relevant points.** In Section 3, we argued that, in contrast to other methods, RHOLS does not prioritize less relevant points that have low probability under the input density $p_{true}(x)$. To investigate this, we subset the CINIC10 dataset; the train set consists of all classes, while the holdout and test set consists of half of the classes. Such a setup could also be useful in some real-world scenarios where one would like to avoid certain training points containing e.g. low-quality, biased, or toxic information by training the irreducible loss model on a clean holdout set.

RHOLS avoids points whose labels do not appear in the validation set (Figure 4). Accordingly, it provides a large speedup. Although irreducible loss selection also avoids irrelevant points, RHOLS selects relevant points that are not yet learned, leading to a greater speedup. RHOLS leverages the implicit task relevant description provided by the holdout data to accelerate model training.

## 5.3 CHEAP IRREDUCIBLE LOSS MODELS

While RHOLS reduces the number training of steps required across several datasets, it requires training an additional model on a separate holdout set, which poses additional computational cost. Here, we examine how to minimize this cost and amortize it across many target model training runs. For practicality, we perform these experiments on the clean benchmark datasets, although we expect RHOLS to speeds up training more on noisy and redundant web scraped data.

**Irreducible loss models can be small and cheap.** While Eq. (6) assumes that the target and irreducible loss model architectures are identical, we hypothesize that this is not necessary. Across sufficiently similar architectures, the irreducible loss is likely lower for points similar to the holdout data, and higher for points with a noisy label (we verify this in Appendix D). None of these require the architectures to be identical. Furthermore, the loss of the target model is used to deprioritize redundant points, and is thus unaffected by the irreducible loss model used.

In Figure 5, we replaced our ResNet-18 irreducible loss model with a small CNN reminiscent of LeNet. It has 21x fewer parameters and requires 29x fewer FLOPs per forward pass than the target ResNet-18. The smaller irreducible loss model accelerates training as much or more than larger model, even though its final accuracy is far lower than the target ResNet-18 (11.5% lower on CIFAR-10, 7% on CIFAR-100, and 8.1% on CINIC-10).

**Irreducible loss models require little holdout data** In Figure 5, we train the irreducible loss model on a small fraction of the available data, $2 - 4$x less data than in our default setting above. To maximize compute savings, we still use the small CNN model. In combination, the computational cost

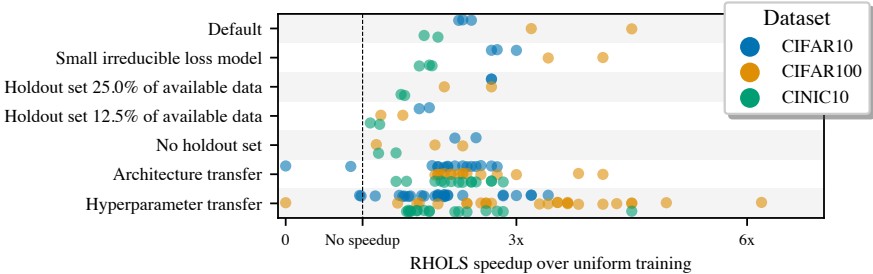

Figure 5: **The irreducible loss model can be small, trained on little data, and reused across target models and hyperparameters.** The x-axis shows after how many fewer epochs RHOLS reaches the highest accuracy that uniform selection reaches during 100 epochs (with $\frac{n_b}{n_B} = 0.1$). *All rows but the first ('Default') use a small CNN as irreducible loss model* (in addition to the other experiment conditions indicated). Each dot represents an experiment with a specific combination of irreducible loss model and target model (multiple seeds per combination shown as separate dots).

is reduced 116x compared to experiments in Section 5.2. The speedup reduces when the irreducible loss model is trained on less data, but RHOLS still speeds up training on each dataset. Note that for web-scraped data, the dataset is often too large to finish even one epoch, meaning that practitioners can afford larger holdout sets.

**Irreducible loss models do not require any holdout data** In Figure 5, we train the irreducible loss model without holdout data. We split the training set into two halves and train a smaller irreducible loss model on each half. Each model computes the irreducible loss for the half of $D$ that it was not trained on. Although we train two models, each model is trained on half the data (assuming equally sized train and validation sets); as such, this approach does not incur additional computational cost.

**Irreducible loss models can be reused to train different target architectures** We find that a single small CNN irreducible loss model accelerates the training of 7 target architectures: VGG11 (with batchnorm), GoogleNet, Resnet34, Resnet50, Densenet121, MobileNet-v2, Inception-v3. On CIFAR-10, RHOLS does not accelerate training of VGG11 (2 seeds), which is also the architecture on which uniform training performs the worst; i.e. RHOLS empirically does not "miss" a good architecture.

**Irreducible loss models can be reused to train many targets in a hyperparameter sweep** We find that a single small CNN accelerates the training of all ResNet-18 target models across a hyperparameter grid search. We vary the batch size (160, 320, 960), learning rate (0.0001, 0.001, 0.01), and weight decay coefficient (0.001, 0.01, 0.1). RHOLS speeds up training compared to uniform on nearly all target hyperparameters. The few settings in which it doesn't speed up training are also settings in which uniform training performs very poorly ($< 30\%$ accuracy on CIFAR-100, $< 80\%$ on CIFAR-10).

## 6 DISCUSSION

With ever larger models and datasets, training times keep low-resourced groups from access to deep learning and well-resourced groups from pushing its frontier. This can already be mitigated by existing methods for parallelization or data selection. However, the benefits are limited by diminishing returns for the former and, as we show, problems with noisy data for the latter. We aim to alleviate these limits by introducing a novel, theoretically grounded selection function. In controlled experiments, we show that its deployment enables substantial speedups on curated data and even larger ones under label noise. However, our approach should be combined with methods in Section 4 for cheap and fast selection with maximal speedups.

## 7 ETHICS STATEMENT

It will be important to understand how subset selection might affect performance on data about minority groups. It is possible that the selection deprioritizes data points on rare groups (since they affect the loss on holdout data less, or prioritizes them late in training once majority groups are already learned.

Since such biases can also stem from the dataset itself (Mehrabi et al., 2021), it should be investigated if our method can remove data biases through the use of an unbiased holdout set. By training the irreducible loss model on unbiased data, we can implicitly specify the task to be learned (see Figure 4). This may be useful for specifying that all groups are equally important to learn.

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
