# OpenReview forum: "Prioritized training on points that are learnable, worth learning, and not yet learned"
_ICLR.cc/2022/Conference — ICLR 2022 Submitted_

### Official Review · Reviewer_DHeZ · 2021-11-01

**Correctness:** 3
**Technical Novelty And Significance:** 3
**Empirical Novelty And Significance:** 2
**Recommendation:** 6
**Confidence:** 4

**Details Of Ethics Concerns:**

It is not clear how selecting a subset of examples during training impacts minority groups. It is known, for example, that related effects (e.g. pruning) can have a disproportionate impact on minorities so that should be looked at as well. It is possible that the model selects typical examples first (by learning general rules that apply to them) and focuses on the minority examples later during training (by memorization). In that case, there would be error disparities between both subpopulations.

**Main Review:**

The paper proposes a technique to speed up training by another trained model as a proxy for how informative examples are. Specifically, starting with the reduction in loss as a criteria, the authors use Bayes theorem to come up with an approximation called the reducible held-out loss (RHOL). Basically, RHOL compares the differences between the current model trained so far and another model trained a separate holdout dataset. A larger difference indicates that the sample is informative and should be selected in the batch.

Overall, I think the method is neat. However, it defeats the original purpose of the work, which is to present a method for accelerating training in the very large data regime. Suppose one has a large dataset with hundreds of millions of examples (as in CLIP or ViT). Using a model trained on a small holdout dataset as a proxy for informativeness is questionable. On the other hand, if the proxy model is trained on a large hold-out dataset, then training is not really accelerated since one has to train two models now. In the experiments, the authors use small datasets only (of CIFAR10/100 size). In addition, the holdout dataset they use is of a comparable size to the original training set (e.g. half of the images in CIFAR10 are used for holdout). In Page 4, the authors do state that their loss rests on the assumption that the holdout set is large when they argue that approximating h(y | x, holdout, D) with h(y |x, holdout) does not have a big impact.

In addition, there is the issue of reserving a labeled holdout set. In CIFAR10, for example, half of the examples are reserved. The authors should compare against the baseline method that is trained on the full data (both training and holdout) since that's really the alternative. I don't think that reserving a large subset of labeled examples and training another model on those is a good solution for "accelerating" training.

Experimentally, the improvement in speed reported in the paper is around a factor of 2. Since another model is trained and a subset of the labeled data is reserved, I don't find the speedup significant enough for this approach to be useful.

Some minor comments:
- The reducible holdout loss criteria is not "information-theoretic" as is claimed in the abstract.
- It is not clear how selecting a subset of examples during training impacts minority groups. It is known, for example, that related effects (e.g. pruning) can have a disproportionate impact on minorities so that should be looked at as well. It is possible that the model selects typical examples first (by learning general rules that apply to them) and focuses on the minority examples later during training (by memorization). In that case, there would be error disparities between both subpopulations.

=========

Post Rebuttal:

I have increased my score based on the new materials added to the paper that addressed some of my concerns.

**Summary Of The Paper:**

The paper proposes a technique to speed up training by using another trained model as a proxy for how informative examples are. The authors conduct experiments on small datasets (e.g. CIFAR10/100) on ResNet-18 to support their claims.

**Summary Of The Review:**

The paper is well-written and easy to read, and the proposed criterion is neat. However, implementing this method in practice for the purpose of speeding up training would require: (1) training another model, and (2) reserving a subset of the labeled data as holdout. I think both requirements constitute serious limitations that defeat the original purpose.

In addition, the experiments were conducted on small datasets only (at least ImageNet should have been included). In those experiments, the holdout data was of a comparable size to the training set and the improvement in speed is only around a factor of 2, which is not significant given the above concerns.

---

> ### Author Response · Authors · 2021-11-23
> **Response part 1/2**
>
> Thank you for these thorough comments! We appreciate that you found that our paper is “well-written and easy to read, and the proposed criterion is neat”.
>
> **Computational and data requirements for training the irreducible loss model**
>
> > Overall, I think the method is neat. However, it defeats the original purpose of the work, which is to present a method for accelerating training in the very large data regime. Suppose one has a large dataset with hundreds of millions of examples (as in CLIP or ViT). Using a model trained on a small holdout dataset as a proxy for informativeness is questionable. On the other hand, if the proxy model is trained on a large hold-out dataset, then training is not really accelerated since one has to train two models now. [... addressed below…]. In Page 4, the authors do state that their loss rests on the assumption that the holdout set is large when they argue that approximating h(y | x, holdout, D) with h(y |x, holdout) does not have a big impact.
>
> > In addition, there is the issue of reserving a labeled holdout set. In CIFAR10, for example, half of the examples are reserved. The authors should compare against the baseline method that is trained on the full data (both training and holdout) since that's really the alternative. I don't think that reserving a large subset of labeled examples and training another model on those is a good solution for "accelerating" training.
>
> Thank you for raising that our original submission had not fully addressed the computational and data requirements of training the irreducible loss model. We have added a new section to the paper that demonstrates that in practice **the irreducible loss model accelerates training even when it has few parameters, low accuracy, and is trained on a small subset of the available data**. Furthermore, **a single irreducible loss model can be used to accelerate training across many architectures and hyperparameter sweeps**. In other words, the cost of training the irreducible loss model is amortised across many runs (and could even be shared between practitioners if they work on the same dataset). In addition, note that in our application context (web scraped data), a large holdout set is often available (see Section 5.0). . Finally, we now report new experiments on Clothing-1M, a web-scraped dataset (see next section of the response, or Figure 3). In these experiments, we train the irreducible loss model on only 5% of the dataset.
>
> This is the new section:
> ### New section: cheap irreducible loss models
>
> **The irreducible loss model can be small, trained on little data, and reused across different target model architectures**
>
> While RHOLS reduces the number training of steps required across several datasets, it requires training an additional model on a separate holdout set, which poses additional computational cost. Here, we examine how to minimize this cost and amortize it across many target model training runs. For practicality, we perform these experiments on the clean benchmark datasets, RHOLS leads to greater speedups on noisy and web-scraped data (Figure 3).
>
> Figure: Figure 5 in paper, or here [https://ibb.co/Rh74P1m](https://ibb.co/Rh74P1m)
>
> **Irreducible loss models can be small and cheap.**
>
> Although our theory section assumes that the target model and irreducible loss model have the same model class, we hypothesize that this is not necessary. [Note to the reviewer: the same considerations also apply to our approximation.]. Across sufficiently similar architectures, the irreducible loss is likely lower for points similar to the holdout data, and higher for points with a noisy label (we verify the latter in Appendix D). None of these require the architectures to be identical. Furthermore, the loss of the target model is used to deprioritize redundant points, and this is unaffected by the irreducible loss model used. We thus experiment with making the irreducible loss model cheaper by using a smaller model and training it on less data.
>
> In the second row of Figure 5, we replaced our ResNet-18 irreducible loss model with a small CNN reminiscent of LeNet. It has 21x fewer parameters and requires 29x fewer FLOPs per forward pass than the target model, ResNet-18. This small irreducible loss model accelerates training as much or more than larger model, even though its final accuracy is far lower than the target ResNet-18 (11.5% lower on CIFAR-10, 7% on CIFAR-100, and 8.1% on CINIC-10).
>
> **We continue in the next comment.**

---

> > ### Author Response · Authors · 2021-11-23
> > **Response part 2/3**
> >
> > We continue the new section and the response here.
> >
> > **Irreducible loss models require little holdout data.**
> >
> > In Figure 5, we train the irreducible loss model on a small fraction of the available data, 2-4x less data than in our default setting above. To maximize compute savings, we still use the small CNN model and train it for 2-4x fewer steps. In combination, **the computational cost of training the irreducible loss model is reduced up to 116X** compared to experiments in Section 5.2. The speedup reduces when the irreducible loss model is trained on less data, but RHOLS still speeds up training on each dataset. Note that for web-scraped data, the dataset size is often too large to finish even one epoch [6-8], meaning that practitioners can work with larger holdout sets.
> >
> > **Irreducible loss models do not require any holdout data.**
> >
> > In row 5, we train the irreducible loss model without holdout data. We split the training set $D$ in two halves and train a (small CNN) irreducible loss model on each half. Each model computes the irreducible loss for the half of D that it was not trained on. Although we train two models, each model is trained on half the data; compared to our default settings, it incurs no additional computational cost.
> >
> > **Irreducible loss models can be reused to train different target architectures.**
> >
> > In row 6, we find that a single small CNN irreducible loss model accelerates the training of 7 new target architectures: VGG11 (with batchnorm), GoogleNet, Resnet34, Resnet50, Densenet121, MobileNet-v2, Inception_v3. On CIFAR-10, RHOLS does not accelerate training of VGG11 (2 seeds), which is also the architecture on which uniform training performs the worst; i.e. RHOLS empirically does not “miss” a good architecture. A single irreducible loss model can thus be reused by many practitioners and researchers or they could simply download the irreducible losses for each point from a repository online.
> >
> > **Irreducible loss models can be reused to train many targets in a hyperparameter sweep.**
> >
> > In row 7, we find that a single small CNN accelerates the training of nearly all ResNet-18 target models across a hyperparameter grid search. We vary the batch size (160, 320, 960), learning rate (0.0001, 0.001, 0.01), and weight decay coefficient (0.001, 0.01, 0.1). RHOLS speeds up training compared to uniform on nearly all target hyperparameters. The few settings in which it doesn’t speed up training are also settings in which uniform training performs very poorly (<30% accuracy on CIFAR-100, <80% on CIFAR-10).
> >
> >
> >
> > ----------------------------
> >
> > ----------------------------
> >
> >
> >
> > ## New large-scale experiment
> >
> > > In the experiments, the authors use small datasets only (of CIFAR10/100 size). In addition, the holdout dataset they use is of a comparable size to the original training set (e.g. half of the images in CIFAR10 are used for holdout).
> >
> >
> > This is a valid point. We report new results on Clothing-1M [9].
> >
> > **New experiment: large web-scraped data**
> >
> > We chose Clothing-1M because
> > * It is large. It has 16x more images, each with 64x more pixels than CIFAR-10/100. This is the same number of datapoints as ILSVRC / ImageNet.
> > * It is the most widely accepted benchmark for noisy labels [10].
> > * It is the highest-ranked image classification dataset on Papers-with-code that has web scraped labels [11], a key application area for our work.
> >
> > Results are shown in the new Figure 3 and in this link:
> >
> > [https://ibb.co/LC23z7V](https://ibb.co/LC23z7V)
> >
> > On Clothing-1M, none of the baselines match uniform random selection, suggesting they are not robust to noise. In contrast, our method reaches the accuracy that uniform selection has after 50 epochs in just 2 epochs and significantly higher final accuracy. Notably, this speedup was possible without any changes to our default hyperparameters or architecture, suggesting that RHOLS is robust and easy to scale. In the experiments shown in Figure 3, the irreducible loss model is trained on a small (5% of training data) but clean holdout set (see experiment section).
> >
> > We continue the response in the next comment.
> >
> > ----------------------------

---

> > > ### Author Response · Authors · 2021-11-23
> > > **Response part 3/3**
> > >
> > > > Experimentally, the improvement in speed reported in the paper is around a factor of 2. Since another model is trained and a subset of the labeled data is reserved, I don't find the speedup significant enough for this approach to be useful.
> > >
> > > We hope that our new section already addresses this point. Additionally, note that speedups are larger on web scraped data, our application area of interest. On Clothing-1M, the speedup is 12.5x. On clean datasets with 10% label noise (a low value compared to typical web scraped data), speedups were 2.34x to 5.15x. Note also that we focus just on evaluating a selection method without tuning our training pipeline for maximal speedups. Our hyperparameters are the default Pytorch settings. Our selectivity hyperparameter is always set to 10%. For example, when we set it to 5%, the speedup on clean CIFAR-10 increases from 3.54x to 10.5x (Appendix F). Finally, we have not used prior techniques that increase speedup, such as stochastic sampling [5].
> > >
> > > > Some minor comments:
> > >
> > > > The reducible holdout loss criteria is not "information-theoretic" as is claimed in the abstract.
> > >
> > > Our selection function is actually the pointwise mutual information (PMI) between the label y and the holdout labels y_ho. We have moved some information-theoretic language to Appendix B to ease the reading.
> > >
> > > > It is not clear how selecting a subset of examples during training impacts minority groups.
> > >
> > > A good point we had not considered. We have added the following ethics statement:
> > >
> > > “It will be important to understand how subset selection might affect performance on data about minority groups. It is possible that the selection deprioritizes data points on rare groups (since they affect the loss on holdout data less, or prioritizes them later on once majority groups are already learned.
> > >
> > > Since such biases can also stem from the dataset itself [1], it should be investigated if our method can remove data biases through the use of an unbiased holdout set. By training the irreducible loss model on unbiased data, we can implicitly specify the task to be learned (see Figure 4). This may be useful for specifying that all groups are equally important to learn.”
> > >
> > > **References**
> > >
> > > [1] Mehrabi, Ninareh, Fred Morstatter, Nripsuta Saxena, Kristina Lerman, and Aram Galstyan. "A survey on bias and fairness in machine learning." ACM Computing Surveys (CSUR) (2021).
> > >
> > > [2] Bottou, Léon, and Yann LeCun. "Large scale online learning." Advances in neural information processing systems 16 (2004).
> > >
> > > [3] Brown, Tom B., Benjamin Mann, Nick Ryder, Melanie Subbiah, Jared Kaplan, Prafulla Dhariwal, Arvind Neelakantan et al. "Language models are few-shot learners." arXiv preprint arXiv:2005.14165 (2020).
> > >
> > > [4] Komatsuzaki, Aran. "One epoch is all you need." arXiv preprint arXiv:1906.06669 (2019).
> > >
> > > [5] Jiang, Angela H., Daniel L-K. Wong, Giulio Zhou, David G. Andersen, Jeffrey Dean, Gregory R. Ganger, Gauri Joshi et al. "Accelerating deep learning by focusing on the biggest losers." arXiv preprint arXiv:1910.00762 (2019).
> > > References:
> > >
> > > [6] Bottou, Léon, and Yann LeCun. "Large scale online learning." _Advances in neural information processing systems_ 16, 2004
> > >
> > > [7] Brown, Tom B., Benjamin Mann, Nick Ryder, Melanie Subbiah, Jared Kaplan, Prafulla Dhariwal, Arvind Neelakantan et al. "Language models are few-shot learners." _arXiv preprint arXiv:2005.14165_, 2020.
> > >
> > > [8] Komatsuzaki, Aran. "One epoch is all you need." _arXiv preprint arXiv:1906.06669_, 2019.
> > >
> > > [9] Tong Xiao, Tian Xia, Yi Yang, Chang Huang, and Xiaogang Wang.  “Learning from massive noisy labeled data for image classification.” In CVPR, 2015.
> > >
> > > [10] Algan, Görkem, and Ilkay Ulusoy. "Image classification with deep learning in the presence of noisy labels: A survey." Knowledge-Based Systems 215, pp. 106771,  2021.
> > >
> > > [11] "Image Classification | Papers With Code." https://paperswithcode.com/task/image-classification. Accessed 20 Nov. 2021.

---

> > > > ### Comment · Reviewer_DHeZ · 2021-11-30
> > > > **New materials are useful**
> > > >
> > > > I think the new materials are quite useful but the narrative of the paper may need to be revised. If we think about the proposed criteria, it says that an example is informative if the current model performs poorly on it while *some* other model performs well. Using the other model as a guide implicitly assumes that the other model is better in some sense (e.g. it is trained on a large holdout dataset as was assumed in the mathematical derivation).
> > > >
> > > > However, the authors show that there is another useful application: transferring knowledge during architecture sweep. When training an architecture (e.g. ResNet50) and training another architecture later (e.g. ResNet101), one can make use of what has been learned by the first architecture to accelerate training in the second. I think this can be quite useful in practice.
> > > >
> > > > Including a large dataset such as Clothing-1M is useful and the results in Figure 3 are interesting. The speed-up seems bigger in the large dataset.
> > > >
> > > > Overall, I am raising my score.
> > > >
> > > > However, there is still the ethics concern.

---

> > > > > ### Author Response · Authors · 2021-11-30
> > > > > **Response**
> > > > >
> > > > > ## Narrative
> > > > >
> > > > > > I think the new materials are quite useful but the narrative of the paper may need to be revised. If we think about the proposed criteria, it says that an example is informative if the current model performs poorly on it while some other model performs well. Using the other model as a guide implicitly assumes that the other model is better in some sense (e.g. it is trained on a large holdout dataset as was assumed in the mathematical derivation).
> > > > >
> > > > > Thank you for your response and for raising your score.
> > > > >
> > > > > Your main remaining concern seems to be the narrative of the paper. While we have already made some changes, we need to continue to improve the narrative as we move for the camera-ready version. Concretely, we need to explain how using an irreducible loss model can accelerate the target model training, even if the irreducible loss model is not already “better” than the target model.
> > > > >
> > > > > Empirically, the irreducible loss model does not have to be better than the target model. **The irreducible loss models in Section 5.3 are much smaller than the target models (>20x fewer parameters, 29x less compute) and reach considerably lower accuracy (11.5% lower on CIFAR-10, 7% on CIFAR-100, and 8.1% on CINIC-10); they still accelerate the target model training considerably.**
> > > > >
> > > > > There are clear intuitions for this observation. For example, take an image of a cat mislabeled as a dog. Even a weak irreducible loss model likely has a higher loss on this mislabeled point than on most other points. This is enough to not select noisy/mislabelled points; the irreducible loss model does not have to outperform the target model (we verify this in Appendix D). Furthermore, the loss of the target model is used to deprioritize redundant points (e.g. multiple repetitions of the same image in web-scraped data), and is thus unaffected by the irreducible loss model used.
> > > > >
> > > > > We do now make these points in the updated manuscript (Sections 3 - Understanding reducible loss, and Section 5.3). However, these sections come quite late in the manuscript. We agree that these points should become a key part of the narrative, and will include them in the abstract and introduction for the camera-ready version.
> > > > >
> > > > > ## Ethics concern
> > > > > > However, there is still the ethics concern.
> > > > >
> > > > > We strongly agree with this comment and believe an interesting direction for future work is how data selection methods, in general, affect minority groups. We note the concern raised by the reviewers (in detail) in our ethics statement. If further ethics acknowledgments or experiments are needed we would be glad to add these in the final paper.
> > > > >
> > > > > Additionally, our approach might also offer a solution to biased training data. By using a curated holdout set, our approach allows the user to specify which groups should be learned. We demonstrate this in Figure 4: by using a holdout set that only contains certain classes, our method selects those classes. If the user creates a holdout set that represents minority groups adequately, it could boost their representation. We also note this in the ethics statement.
> > > > >
> > > > > &nbsp;
> > > > >
> > > > > Have we addressed your remaining concerns with this clarification?

---

### Official Review · Reviewer_Yofy · 2021-11-01

**Correctness:** 3
**Technical Novelty And Significance:** 3
**Empirical Novelty And Significance:** 3
**Recommendation:** 6
**Confidence:** 4

**Main Review:**

Strength:

The paper presents a method for speeding up the training of DNN models. The problem addressed in the paper is important given the long training demand by data-centric applications.


Weakness:

1. The claim about training speed-up is evaluated on smaller datasets, whereas the targeted problem appears real while training with huge data. Therefore, it will be of interest to see the performance of the proposed method on bigger datasets.

2. A theoretical justification about training speed up is missing.

3. More reasoning is required for the approximation used in Equation (5). How to ensure that Dt does not cover the relevant (most significant) data points. There may be chances that more decisive data samples are there in Dt  therefore this approximation may not hold true.

4. Training speedup is measured in terms of a number of steps but what does a step consist of, is not clear. More discussion is required about Figure 2.

5. Which parameters are tuned by Algorithm-1 is not clear.

6. It will be of great interest to see the comparative performance of the proposed method in terms of wall-clock time in a constant experimental setting.

**Summary Of The Paper:**

The paper presents a method for selecting the most appropriate data points and using them for training to realize the speedup in training. The method proposes the use of reducible loss using a novel selection function named as Predictive Information Content. The paper claims that proposed RHOLS function helps in reaching the benchmark accuracy in fewer steps.

**Summary Of The Review:**

It is a good work with good technical contributions on an important problem.
Authors need to rectify / address the outlined concerns above.

---

> ### Author Response · Authors · 2021-11-23
> **Response part 1/2**
>
> We appreciate that you found the paper “a good work with good technical contributions on an important problem”. Thank you for the thorough comments! We will address them below in turn.
>
> **New experiment: large-scale web-scraped data**
>
> > 1. The claim about training speed-up is evaluated on smaller datasets, whereas the targeted problem appears real while training with huge data. Therefore, it will be of interest to see the performance of the proposed method on bigger datasets.
>
> This is a valid point. We report new results on Clothing-1M [5].
>
> We chose Clothing-1M because
> * It is large. It has 16x more images, each with 64x more pixels than CIFAR-10/100. This is the same number of datapoints as ILSVRC / ImageNet.
> * It is the most widely accepted benchmark for noisy labels [6].
> * It is the highest-ranked image classification dataset on Papers-with-code that has web scraped labels [7], a key application area for our work.
>
> Results are shown in the new Figure 3 and in this link:
>
> [https://ibb.co/LC23z7V](https://ibb.co/LC23z7V)
>
> On Clothing-1M, none of the baselines match uniform random selection, suggesting they are not robust to noise. In contrast, **our method reaches the accuracy that uniform selection has after 50 epochs in just 2 epochs and significantly higher final accuracy**. Notably, this speedup was possible without any changes to our default hyperparameters or architecture, suggesting that RHOLS is robust and easy to scale. In the experiments shown in Figure 3, the irreducible loss model is trained on a small (5% of training data) but clean holdout set (see experiment section).
>
> > 2. A theoretical justification about training speed up is missing.
>
> We are uncertain about what type of theoretical support the reviewer seeks—additional clarification would be beneficial. To summarize, we prove that, under appropriate assumptions, our exact method (eq. 6) selects the next point for training that most reduces the (joint cross entropy) loss on holdout data. In other words, it maximizes the Shannon information gained about the holdout labels. The derivation is shown in Appendix A. If our result is not clearly presented, we are happy to improve it.
>
> Compared to other results in the literature, we have comparable or stronger theoretical results. In particular, no closely related works we know prove speedups in convergence rate. Instead, these works, including our own, prove speedups that are greedy, i.e. they minimize/maximize some metric for the next step. For instance, [2] minimizes the variance of the next gradient update and [3] maximizes information gained about the true parameters. Our result is stronger in that we select the points for training that minimizes a metric directly relevant to practitioners: the holdout loss. This has intuitively useful properties such as deprioritizing noisy points and prioritizing ones that are close to holdout data. The result would be stronger if our speedup was not greedy. However, this would require strong assumptions on the loss landscape (such as convexity) or on the dataset [1] (e.g. a speedup is not possible if all data points are identical; we would have to assume this and similar cases do not hold).
>
>
>
> > 3. More reasoning is required for the approximation used in Equation (5). How to ensure that Dt does not cover the relevant (most significant) data points. There may be chances that more decisive data samples are there in Dt therefore this approximation may not hold true.
>
> We first explain why our method works without updating on Dt and then on why the particular situation described by the reviewer is likely not a problem.
>
> **The approximation need not be accurate to accelerate model training.** Our method accelerates training by avoiding noisy and redundant points, and preferring points close to the holdout data. Redundant points likely have low loss under the target model, and thus will be deprioritised. Noisy points likely have high (generalisation) loss under the irreducible loss model, _even if the model is not updated on the training data_ (Figure 1B). Points closer to the holdout data likely have lower loss for the model trained on the holdout data, and will thus be preferred. As a result, we expect RHOLS to accelerate training even if the irreducible loss model is updated on the training data. Extensive experiments provide further evidence that this is the case. We have added these experiments and the above reasoning in Section 5.
>
> **We continue the response in the next comment**

---

> > ### Author Response · Authors · 2021-11-23
> > **Response part 2/2**
> >
> > We continue from the previous comment here.
> >
> > Furthermore, the reviewer asks if it is problematic to not update on Dt when Dt contains points x’ that are relevant (i.e. close) to a new sample x. This is not problematic when the holdout set also contains points relevant to x. To illustrate, suppose we want to select an important point x, and x is close to holdout points x’’ in x_ho AND close to some training points. All else equal, the irreducible losses h(y | x, x_ho, y_ho, Dt) and h(y | x, x_ho, y_ho) will be similar since x is close both to points x’’ in x_ho and x’ in Dt. Therefore, x is similarly likely to be selected for training with or without the approximation. Thus it is not a problem that Dt contains samples x’’ that are highly ‘relevant’ to x. Finally, an ablation for our approximation is included in Appendix C, finding no difference in training speed.
> >
> >
> >
> > > 4. Training speedup is measured in terms of a number of steps but what does a step consist of, is not clear. More discussion is required about Figure 2. 5. _Which parameters are tuned by Algorithm-1 is not clear. 6. It will be of great interest to see the comparative performance of the proposed method in terms of wall-clock time in a constant experimental setting.
> >
> > For these three comments, we have clarified that a step in Figure 2 corresponds to lines 6-11 in Algorithm 1. For our baselines, these lines are different but also consist of forward pass, selection, backward pass, gradient update, and require the same amount of computation. We evaluate methods in terms of the number of steps (not wallclock time) required for the reasons given in sections 3 (final paragraph) and 5.2. Furthermore, we have tried to clarify that the updated parameter is theta_t, although we are unsure in which line the clarification is needed.
> >
> > The response to other reviewers may also be of interest. We have added extensive experiments on compute and data-efficient selection.
> >
> >
> > **References**
> >
> > [1] Hanneke, Steve. "A bound on the label complexity of agnostic active learning." Proceedings of the 24th international conference on Machine learning, pp. 353-360. 2007.
> >
> > [2] Katharopoulos, Angelos, and François Fleuret. "Not all samples are created equal: Deep learning with importance sampling." International conference on machine learning. PMLR, 2018.
> >
> > [3] Houlsby, Neil, et al. "Bayesian active learning for classification and preference learning." arXiv preprint arXiv:1112.5745, 2011.
> >
> > [4] Gaelle Loosli, Stephane Canu, and Leon Bottou. “Training invariant support vector machines using selective sampling.” In Leon Bottou, Olivier Chapelle, Dennis DeCoste, and Jason Weston (eds.), Large Scale Kernel Machines, pp. 301–320. MIT Press, Cambridge, MA., 2007.
> >
> > [5] Tong Xiao, Tian Xia, Yi Yang, Chang Huang, and Xiaogang Wang.  “Learning from massive noisy labeled data for image classification.” In CVPR, 2015.
> >
> > [6] Algan, Görkem, and Ilkay Ulusoy. "Image classification with deep learning in the presence of noisy labels: A survey." Knowledge-Based Systems 215, pp. 106771,  2021.
> >
> > [7] "Image Classification | Papers With Code." https://paperswithcode.com/task/image-classification. Accessed 20 Nov. 2021.

---

### Official Review · Reviewer_BBTj · 2021-11-02

**Correctness:** 4
**Technical Novelty And Significance:** 2
**Empirical Novelty And Significance:** 3
**Recommendation:** 5
**Confidence:** 4

**Main Review:**

### Strengths

1. In general, the paper is well-written, with the main ideas outlined clearly. The approach is well-motivated, backed by information-theoretical reasonings, and seems to be built upon established literature.
2. One good contribution of the paper is the illustrations of why hard examples may be inappropriate in some application contexts.
3. The simulation part of the manuscript was carefully constructed with appropriate experiments to support the main findings. The visualizations of the results are very appealing.
4. The derivation of the PIC (and to some extent, the problem considered in this work) is motivated by developments in active learning but is adapted in a novel way. (This might be either a strength or a weakness, depending on the viewpoints, as well as the perceived importance of the proposed problem).

### Weakness

1. As stated above, the framework of the problem is basically an easier version of a classical active learning problem. In active learning, examples' labels are requested within a budget; in the context of this manuscript, they are already available on both the train set and the hold-out dataset. Aside from the novelty of the applications, other aspects of the approach resemble classical discussions in active learning. These include the contrast between hard/easy samples or aggressive/mellow learning, and the central insight that hard examples might be noisy (which is expected from the active learning literature, many of which are referenced in this work).
2. The manuscript is a practical demonstration of an idea to prioritize samples for learning and doesn't have strong theoretical supports (that usually accompany active learning methods).

**Summary Of The Paper:**

This paper introduces a new technique to select a sequence of training points for faster model training. The newly derived method is tractable, can improve efficiency through selections of examples that are neither too hard nor too easy, and is robust to noise.

**Summary Of The Review:**

Overall, my vote for the paper is a (weak) reject. I enjoy reading the paper and think it has some good points and the discussions are meaningful. On the other hand, I think the technical contribution of the work is limited.

---

> ### Author Response · Authors · 2021-11-23
> **Response part 1/2**
>
> Thank you for these insightful comments. We appreciate that you found our work “well-written”, “well-motivated”, and “backed by information-theoretical reasonings”, as well as a “good contribution” that is “motivated by active learning but adapted in a novel way”, with “carefully constructed” experiments and “very appealing” visualization of the results. Below, we respond to the three concerns raised.
>
>
> ### Comment 1: significance
>
> > Depending on [...] the perceived importance of the proposed problem. [...] As stated above, the framework of the problem is basically an easier version of a classical active learning problem. In active learning, examples' labels are requested within a budget; in the context of this manuscript, they are already available on both the train set and the hold-out dataset.
>
> **Accelerating model training is an important problem**
>
> We believe that accelerating the training of deep learning models is significant and impactful [9, 10]. Neural network training, especially for large networks, not only costs valuable time, but also has high energy, carbon, and thus social costs [8]. What makes these issues even more significant is that they affect a large number of ML projects [9, 10] e.g. through numerous iterations of hyperparameter selection and algorithm design.
>
> Furthermore, we highlight that our subfields—Online Batch Selection (OBS) and Core-Set Selection, which are commonly seen as the labeled versions of active learning (AL)—are active research fields with complex and important papers [9, 10]. However, we agree that we have not sufficiently highlighted the differences to AL. We will add this discussion to the paper. The evaluation of this paper should not depend on whether these fields can be seen as “an easier version of active learning”. The problem poses additional constraints not usually faced in AL: AL methods rely on retraining the model from scratch or ensembling forward passes for selection, which is not compatible with the (much) tighter computational limits we face. Furthermore, our method must achieve high speedups even when acquiring very large numbers of points. Finally, in our newly added experiments (see below), we find that AL baselines in the online selection setting underperform despite using more computation.
>
>
> ### Comment 2: novelty
>
> > Aside from the novelty of the applications, other aspects of the approach resemble classical discussions in active learning. These include the contrast between hard/easy samples or aggressive/mellow learning, and the central insight that hard examples might be noisy (which is expected from the active learning literature, many of which are referenced in this work).
>
> **Connecting online batch selection to active learning promises greater understanding and performant algorithms**
>
> We agree with the reviewer that “the framework of the problem” is closely related to active learning. Thank you for pointing to these important connections. **Revealing the relationship between active learning and accelerating model training is actually an important benefit of our work.** Prior work on online batch selection and web-scraped data has not appreciated that we can use ideas from active learning to select points—see references [1-4]. This relationship allows us to take inspiration from classical discussions within the active learning subfield, and thus **reveal underappreciated failure models of common online batch selection techniques**. One such example is that choosing hard examples can be inappropriate in important application contexts such as web-scraped data. The reviewer describes this as a “good contribution”. Further, revealing connections between active learning and online batch selection empowers us to develop novel, performant algorithms that exhibit strong efficiency gains in our experiments.
>
> Indeed, knowledge transfer has been mainly in the other direction, e.g. [11, ICLR 2018] which applied well-known ideas from the Core-Set literature as novel contributions to active learning, showing the value of transferring such knowledge.
>
> Further clarification would be helpful here as we are uncertain which aspects of our work appear in prior work. If specific prior works detract from our novelty we would be happy to compare our work to these.
>
> **We continue the response in the next comment**

---

> > ### Author Response · Authors · 2021-11-23
> > **Response part 2/2**
> >
> > Here, we continue the response from the previous comment.
> >
> > **New experiments: training in fewer steps than active learning baselines**
> >
> > Despite the similarities, we **cannot naively apply ideas from AL.** We have added experiments demonstrating that our method trains in fewer steps than active learning baselines that are also designed to avoid noisy points. The first baseline is BALD, a popular and principled Bayesian AL method that avoids noisy points [5, 7] but requires many more forward passes than our method. The other baselines are predictive entropy, conditional entropy, and loss minus conditional entropy.
> >
> > Our method requires significantly fewer steps than active learning baselines:
> >
> > [https://ibb.co/nChZr6w](https://ibb.co/nChZr6w)  (description in Appendix G)
> >
> > Note that the “loss minus conditional entropy” baseline is similar to our method except using the conditional entropy to remove noisy points, instead of the irreducible loss. With only this change, training substantially slows down, suggesting that our irreducible loss method is more effective at removing noisy points.
> >
> > **Our contributions may be of interest to active learning community**
> >
> > Our work also provides insights that may be of interest to the active learning community. We show how to use a holdout dataset that implicitly defines which points are relevant for the task we wish to solve, and which points are not. We demonstrate that our method chooses relevant points—see Figure 4. The relevance and novelty of this idea to AL is reflected in the fact that concurrent work demonstrates it for active learning [6]. To our knowledge, this insight is novel in active learning (and in online batch selection). We have cited [6] to compare our methods but we stress that this paper is concurrent to ours. Indeed, it was published in the same workshop as our work and does not detract from the novelty of our insight.
> >
> > Furthermore, although prior work in AL has considered the general ideas of avoiding noise and redundancy, we contribute novel theory showing how to find the right balance between the competing goals of avoiding noise, avoiding redundancy, and seeking relevance to the holdout set. Satisfyingly, the resulting method (eq. 4) finds the next point for training that will most reduce the holdout loss.
> >
> > ---------------
> >
> >
> > ### Comment 3: Theoretical support
> >
> > > The manuscript is a practical demonstration of an idea to prioritize samples for learning and doesn't have strong theoretical supports (that usually accompany active learning methods).
> >
> > We are uncertain about what type of theoretical support the reviewer seeks—additional clarification would be beneficial. Our approach shows large speedups compared to recent baselines and is developed, as the reviewer notes, by grounding our problem setting in a principled, information theoretic framework.
> >
> > We prove that, under appropriate assumptions, our exact method selects the next point for training that most reduces the (joint cross entropy) loss on holdout data. In other words, it maximizes the information gained about the holdout labels. The derivation is shown in Appendix A.
> >
> > If the reviewer has concerns about our theoretical derivations, or our presentation thereof, we are happy to address this.
> >
> > Our derivation is similar to that for BALD [5], a classic result in Bayesian active learning. Our result is stronger in two ways: 1) BALD maximizes the information gained in expectation whereas we maximize the actual information gained and 2) BALD maximizes information about the parameters as a proxy for minimizing the loss on holdout data, whereas we minimize the holdout loss directly. This theoretical result is backed by empirical insight showing that our method selects points that are non-noisy, non-redundant, and relevant to the holdout data.
> >
> > Our main assumptions are that we acquire one point at a time and update the model using Bayes' rule. These assumptions are standard in Bayesian AL and are also needed in the BALD paper. Additionally, we use an approximation (eq. 5) that is verified empirically (see Appendix D). If anything was unclear in the paper, we would be glad to clarify.
> >
> > Finally, the response to other reviewers may be of interest. We have added many new experiments that include a larger dataset, increased speedups, and compute-efficient selection.
> >
> > References are in the next comment.

---

> > > ### Author Response · Authors · 2021-11-23
> > > **References**
> > >
> > > **[1] Jiang, Angela H., et al. "Accelerating deep learning by focusing on the biggest losers." _arXiv preprint arXiv:1910.00762_, 2019.**
> > >
> > > **[2] Loshchilov, Ilya, and Frank Hutter. "Online batch selection for faster training of neural networks." _arXiv preprint arXiv:1511.06343_, 2015.**
> > >
> > > **[3] Kawaguchi, Kenji, and Haihao Lu. "Ordered SGD: A new stochastic optimization framework for empirical risk minimization." _International Conference on Artificial Intelligence and Statistics_. PMLR, 2020.**
> > >
> > > **[4] Katharopoulos, Angelos, and François Fleuret. "Not all samples are created equal: Deep learning with importance sampling." _International conference on machine learning_. PMLR, 2018.**
> > >
> > > **[5] Houlsby, Neil, et al. "Bayesian active learning for classification and preference learning." _arXiv preprint arXiv:1112.5745_, 2011.**
> > >
> > > **[6] Andreas Kirsch, Tom Rainforth, and Yarin Gal. Active learning under pool set distribution shift and noisy data. CoRR, abs/2106.11719, 2021. URL [https://arxiv.org/abs/2106.11719](https://arxiv.org/abs/2106.11719)**
> > >
> > > **[7] Gal, Yarin, Riashat Islam, and Zoubin Ghahramani. "Deep Bayesian active learning with image data." _International Conference on Machine Learning_, pp. 1183-1192. PMLR, 2017.**
> > >
> > > **[8] Dhar, Payal. "The carbon impact of artificial intelligence." _Nature Machine Intelligence_ 2(8), pp.423-425, 2020.**
> > >
> > > **[9] Menghani, Gaurav. "Efficient Deep Learning: A Survey on Making Deep Learning Models Smaller, Faster, and Better." _arXiv preprint arXiv:2106.08962_, 2021.**
> > >
> > > **[10] Feldman, Dan. "Core-sets: Updated survey." _Sampling Techniques for Supervised or Unsupervised Tasks_ 23-44, 2020.**
> > >
> > > **[11] Sener, Ozan, and Silvio Savarese. "Active learning for convolutional neural networks: A core-set approach." _arXiv preprint arXiv:1708.00489_, 2017.**

---

> > > ### Comment · Reviewer_BBTj · 2021-11-28
> > > **thanks for the revision**
> > >
> > >
> > > I want to thank the author's for the addition of the new experiment and for carefully addressing my concern. However, I felt that the two weaknesses of the paper, most importantly about the novelty and technical contributions of the work in comparison with its counterpart of active learning, are still there.
> > >
> > > Some references (requested by the author) could be:
> > >
> > > 1. On the contrast between hard/easy samples or aggressive/mellow learning: see for example,
> > >
> > > Dasgupta, Sanjoy. "Two faces of active learning." Theoretical computer science 412.19 (2011): 1767-1781.
> > >
> > > 2. On theory of active learning: Theoretical analyses can be obtained in two manners: sample complexity, or (sub-)optimality given fixed budget:
> > >
> > > Cuong, Nguyen Viet, et al. "Active learning for probabilistic hypotheses using the maximum Gibbs error criterion." Advances in Neural Information Processing Systems 26 (NIPS 2013) (2013): 1457-1465.
> > >
> > > Golovin, Daniel, and Andreas Krause. "Adaptive submodularity: Theory and applications in active learning and stochastic optimization." Journal of Artificial Intelligence Research 42 (2011): 427-486.
> > >
> > > Hanneke, Steve. Theoretical foundations of active learning. Carnegie Mellon University, 2009.

---

> > > > ### Author Response · Authors · 2021-11-29
> > > > **Thank you for active learning references**
> > > >
> > > > Thank you for taking the time to provide active learning references. We want to clarify why we believe our work is novel despite these.
> > > >
> > > > For context, note that all other reviewers have pointed out the novelty of our contributions. To quote: “the proposed method is novel” and “conceptually interesting”, with “a novel selection function”, and “good technical contributions”. The only concerns were about practicality (which we have addressed). The positive evaluation of our method’s novelty reflects that our contributions are novel in our field (online batch and Coreset selection) which is not affected by related ideas in another field (active learning).
> > > >
> > > > Our selection function consists of two components: the holdout loss computed by a model trained on a holdout set, and the loss of the target model. None of these have to our knowledge appeared in prior work on active learning. Indeed, active learning methods cannot straight-forwardly be applied to our problem. Our method significantly outperforms active learning baselines.
> > > >
> > > > Many papers take inspiration from a related field, without simply copying the field’s methods, to make important advances. We believe these papers are published for good reason. Well-known examples similar to our work include: 1) Prioritized Experience Replay [1] which combined loss-based selection with importance sampling and applied it to reinforcement learning. And Sener et al. [2] which transferred insights from the Coreset literature to active learning. Finally, BALD [3] which uses the well-known information gain [4] but computes it more cheaply.
> > > >
> > > > We believe it is important to transfer and adapt insights that are not appreciated in one field because different fields come with different constraints. In our case, the selection function must be cheaper to compute than in active learning and must work without retraining the model after each acquisition step.
> > > >
> > > > In addition, **a key contribution of our work is not borrowed from active learning**: we select points that are relevant to the holdout set. This contribution has both empirical and theoretical support (Figure 4 and Section 3).
> > > >
> > > > A further important criterion is empirical novelty. We showed that our irreducible loss can remove points with noisy labels more effectively than active learning baselines, and that our method outperforms active learning baselines by a significant margin.
> > > >
> > > > Regarding theory, note that the theory in the cited papers does not apply to deep learning. It also does not apply to online batch selection. In online batch selection for deep learning, one cannot apply the same theoretical standard that can be satisfied for simple methods on small-scale problems. As mentioned above, the theory we give conforms to the same or stronger standards than principled active learning methods (which do not satisfy adaptive submodularity) such as BALD. This already exceeds the theoretical justification present in papers on deep learning.
> > > >
> > > >
> > > > -------------------------------------------------------------
> > > >
> > > >
> > > >
> > > > **References**
> > > >
> > > > _[1] Schaul, Tom, John Quan, Ioannis Antonoglou, and David Silver. "Prioritized experience replay." arXiv preprint arXiv:1511.05952 (2015)._
> > > >
> > > > _[2] Sener, Ozan, and Silvio Savarese. "Active learning for convolutional neural networks: A core-set approach." arXiv preprint arXiv:1708.00489, 2017._
> > > >
> > > > _[3] Houlsby, Neil, Ferenc Huszár, Zoubin Ghahramani, and Máté Lengyel. "Bayesian active learning for classification and preference learning." arXiv preprint arXiv:1112.5745 (2011)._
> > > >
> > > > _[4] MacKay, David JC, and David JC Mac Kay. Information theory, inference and learning algorithms. Cambridge university press, 2003._

---

### Official Review · Reviewer_NdhY · 2021-11-03

**Correctness:** 2
**Technical Novelty And Significance:** 3
**Empirical Novelty And Significance:** 2
**Recommendation:** 5
**Confidence:** 4

**Main Review:**

While the proposed is conceptually interesting, I have major concerns with the practicality of the proposed method. By assuming an irreducible loss model trained on hold-out the set, authors need access to an already trained model on a separate IID dataset. This seems like begging the task at hand. In particular, to train a model on some training set, Algorithm 1 (RHOLS) assumes access to a model trained on some hold-out set. Then the proposed criterion uses this irreducible model and the training model to identify/select a batch of examples at every iteration.

Other concerns:
- Approximation from 4 to 6 might not be accurate if the hold-out set is comparable or smaller than the training data. If the hold-out set is large then why would one not use that for training the model in the first place?
- It would be interesting to add some discussion on how is the irreducible model is selected. Some ablation experiments showing the differences from equation 4 to equation 6 can be insightful.

I am open to changing my score if there is any misunderstandings with my interpretation.

**Summary Of The Paper:**

The authors propose a reducible held-out loss selection method for faster model training. Their method selects a sequence of training points that are referred to as "just right" for training.  The main methodological contribution of the paper is an information-theoretic selection criterion Predictive Information Content (PIC) to choose training points. Authors emprically show efficacy of their selection criterion over previously proposed baselines.

**Summary Of The Review:**

Overall, while the proposed method is novel, there is some serious concern with the practicality of the RHOLS method because it assumes an already trained model on a hold-out set to train another model on training set.

---

> ### Author Response · Authors · 2021-11-23
> **Response part 1/2**
>
> We appreciate that you found our method “novel” and “conceptually interesting”. Thank you for the feedback on practicality.
>
> **Computational and data requirements of training the irreducible loss model**
>
> >  By assuming an irreducible loss model trained on hold-out the set, authors need access to an already trained model on a separate IID dataset. This seems like begging the task at hand.
>
> > Other concerns:
> Approximation from 4 to 6 might not be accurate if the hold-out set is comparable or smaller than the training data. If the hold-out set is large then why would one not use that for training the model in the first place?
>
> Thank you for highlighting the computational and data requirements of training the irreducible loss model. We include the following new section with over 100 experiments on 3 datasets that we believe addresses these concerns convincingly:
>
> ### New section: cheap irreducible loss models
>
> **The irreducible loss model can be small, trained on little data, and reused across different target model architectures**
>
> While RHOLS reduces the number training of steps required across several datasets, it requires training an additional model on a separate holdout set, which poses additional computational cost. Here, we examine how to minimize this cost and amortize it across many target model training runs. For practicality, we perform these experiments on the clean benchmark datasets, although RHOLS speeds up training more on noisy or web-scraped data (Figure 3).
>
> [Note to reviewer: we changed this paragraph for your benefit to focus on the approximation. The same arguments apply to using an irreducible loss of different size and architecture than the target model.] Although our theory section assumes that the irreducible loss model is trained on the training data Dt (in addition to the holdout data), our approximation (eq. 6) skips this step. The accuracy of our approximation may suffer from using a small holdout set. However, we hypothesize that such accuracy is not needed to fulfill the two purposes of using an irreducible loss model: 1) the irreducible loss is likely lower for points similar to the holdout data, and 2) higher for points with a noisy label (this is shown empirically in Appendix D). None of these require updating the model on Dt. Furthermore, we use the loss of the target model to deprioritize redundant points; this loss is unaffected by updating the irreducible loss model on Dt.
>
> Figure: Figure 5 in paper, or here [https://ibb.co/Rh74P1m](https://ibb.co/Rh74P1m)
>
> **Irreducible loss models can be small and cheap.**
>
> In the second row of Figure 5, we replaced our ResNet-18 irreducible loss model with a small CNN reminiscent of LeNet. It has 21x fewer parameters and requires 29x fewer FLOPs per forward pass than the target model, ResNet-18. This small irreducible loss model accelerates training as much or more than larger model, even though its final accuracy is far lower than the target ResNet-18 (11.5% lower on CIFAR-10, 7% on CIFAR-100, and 8.1% on CINIC-10).
>
> **Irreducible loss models require little holdout data.**
>
> In Figure 5, we train the irreducible loss model on a small fraction of the available data, 2-4x less data than in our default setting above. To maximize compute savings, we still use the small CNN model and train it for 2-4x fewer steps. In combination, **the computational cost of training the irreducible loss model is reduced up to 116X** compared to experiments in Section 5.2. The speedup reduces when the irreducible loss model is trained on less data, but RHOLS still speeds up training on each dataset. Note that for web-scraped data, the dataset size is often too large to finish even one epoch [5-7], meaning that practitioners can work with larger holdout sets.
>
> **Irreducible loss models do not require any holdout data.**
>
> In row 5, we train the irreducible loss model without holdout data. We split the training set $D$ in two halves and train a (small CNN) irreducible loss model on each half. Each model computes the irreducible loss for the half of $D$ that it was not trained on. Although we train two models, each model is trained on half the data; compared to our default settings, it incurs no additional computational cost.
>
> **Irreducible loss models can be reused to train different target architectures.**
>
> In row 6, we find that a single small CNN irreducible loss model accelerates the training of 7 new target architectures: VGG11 (with batchnorm), GoogleNet, Resnet34, Resnet50, Densenet121, MobileNet-v2, Inception_v3. On CIFAR-10, RHOLS does not accelerate training of VGG11 (2 seeds), which is also the architecture on which uniform training performs the worst; i.e. RHOLS empirically does not “miss” a good architecture. A single irreducible loss model can thus be reused by many practitioners and researchers or they could simply download the irreducible losses for each point from a repository online.
>
> **We continue in the next comment.**

---

> > ### Author Response · Authors · 2021-11-23
> > **Response part 2/2**
> >
> > (We continue the new section and the response here.)
> >
> > **Irreducible loss models can be reused to train many targets in a hyperparameter sweep.**
> >
> > In row 7, we find that a single small CNN accelerates the training of nearly all ResNet-18 target models across a hyperparameter grid search. We vary the batch size (160, 320, 960), learning rate (0.0001, 0.001, 0.01), and weight decay coefficient (0.001, 0.01, 0.1). RHOLS speeds up training compared to uniform on nearly all target hyperparameters. The few settings in which it doesn’t speed up training are also settings in which uniform training performs very poorly (<30% accuracy on CIFAR-100, <80% on CIFAR-10).
> >
> >
> > End of section.
> >
> > ----------------------------
> >
> >
> > Further, large holdout sets are often available for the applications we have in mind, such as when computation and time are bottlenecks but data is abundant (see [3]). This is common e.g. for web scraped data where state-of-the-art performance is often reached in less than half of one epoch [1,2]. In such scenarios, there is no downside to setting aside holdout data.
> >
> > Finally, we now report new experiments on Clothing-1M, a large web-scraped dataset (see response to R4, or Figure 3). In these experiments, we reach 12.5x speedup and **train the irreducible loss model on only 5% of the dataset** (with clean labels).
> >
> > > It would be interesting to add some discussion on how is the irreducible model is selected. Some ablation experiments showing the differences from equation 4 to equation 6 can be insightful.
> >
> > We performed no hyperparameter selection on the irreducible loss model. However, we select the checkpoint of the irreducible loss model that has the lowest loss on unseen data, not the highest accuracy. We have now clarified this in Section 5. We select for low loss because our method uses the irreducible loss, not accuracy. This choice also saves computation since the lowest held-out loss is typically achieved early in training. Finally, an ablation of our approximation on MNIST-8M [4] is included in Appendix C, finding no difference in training speed or accuracy.
> >
> > **References:**
> >
> > [1] Brown, Tom B., Benjamin Mann, Nick Ryder, Melanie Subbiah, Jared Kaplan, Prafulla Dhariwal, Arvind Neelakantan et al. "Language models are few-shot learners." arXiv preprint arXiv:2005.14165, 2020.
> >
> > [2] Komatsuzaki, Aran. "One epoch is all you need." arXiv preprint arXiv:1906.06669, 2019.
> >
> > [3] Bottou, Léon, and Yann LeCun. "Large scale online learning." Advances in neural
> > information processing systems 16, 2004
> >
> > [4] Gaelle Loosli, Stephane Canu, and Leon Bottou. “Training invariant support vector machines using selective sampling.” In Leon Bottou, Olivier Chapelle, Dennis DeCoste, and Jason Weston (eds.), Large Scale Kernel Machines, pp. 301–320. MIT Press, Cambridge, MA., 2007.
> >
> > [5] Bottou, Léon, and Yann LeCun. "Large scale online learning." _Advances in neural information processing systems_ 16, 2004
> >
> > [6] Brown, Tom B., Benjamin Mann, Nick Ryder, Melanie Subbiah, Jared Kaplan, Prafulla Dhariwal, Arvind Neelakantan et al. "Language models are few-shot learners." _arXiv preprint arXiv:2005.14165_, 2020.
> >
> > [7] Komatsuzaki, Aran. "One epoch is all you need." _arXiv preprint arXiv:1906.06669_, 2019.

---

> > > ### Comment · Reviewer_NdhY · 2021-11-30
> > > **Response**
> > >
> > > I thank the authors for the detailed clarification and updated experiments.
> > >
> > > I agree with the speed-up argument on very large datasets and applications in architecture search. Hence I am raising my score. However, I think the narrative can be improved a lot in the introduction to motivate the method. I am also unable to identify all the added changes as they are not marked or highlighted in the updated draft.

---

> > > > ### Author Response · Authors · 2021-11-30
> > > > **Response**
> > > >
> > > > Thank you for considering our response and raising your score. Please find the updated manuscript with highlighted changes [here](https://www.dropbox.com/s/l66q8x1srp0z1a8/ICLR2021_CurricuLM_Paper-3.pdf?dl=0). We have changed a lot; the main changes are summarised at the bottom of this response.
> > > >
> > > > Your main remaining concern seems to be the narrative in the introduction. Concretely, we need to explain how training two models is supposed to be faster than training one. There are 2 main reasons:
> > > >
> > > > 1.) As highlighted by you, the irreducible loss model can be reused across different target architectures and hyperparameter settings, so its training cost can be amortized across many target model training runs (Section 5.3).
> > > >
> > > > 2.) We show that the irreducible loss model can be small and trained on little (or no) additional data, meaning it poses little additional cost. The irreducible loss models in Section 5.3 are much smaller than the target models (>20x fewer parameters, 29x less compute) and reach considerably lower accuracy (11.5% lower on CIFAR-10, 7% on CIFAR-100, and 8.1% on CINIC-10); they still accelerate the target model training considerably. The irreducible loss model for the Clothing-1M dataset was trained on only 5% of the training data.
> > > >
> > > > Both 1.) and 2.) can be explained by considering the key properties of our methods: it avoids mislabelled points, redundant points, and points that are not relevant for the holdout set. None of these require the architectures to be identical. As an example, take an image of a cat mislabeled as a dog. Even a weak irreducible loss model likely has a higher loss on this mislabeled point than on most other points. This is enough to not select noisy/mislabelled point (we verify this in Appendix D). Across sufficiently similar architectures, the irreducible loss is also likely lower for points similar to the holdout data. Furthermore, the loss of the target model is used to deprioritize redundant points (e.g. multiple repetition of the same image in web-scraped data), and is thus unaffected by the irreducible loss model used.
> > > >
> > > > We do now make these arguments in the updated manuscript (Sections 3 - Understanding reducible loss, and Section 5.3). However, these sections come quite late in the manuscript and we agree that these arguments should become a key part of the narrative. We will include it in the abstract and introduction. A potential section to add to the introduction in the camera-ready version:
> > > > "A weak irreducible loss model can often be obtained cheaply (Section 5.3). Practitioners can train a small and cheap model. Or they can use a model from a previous run that had suboptimal hyperparameters or a suboptimal architecture. For datasets that other practitioners use, the practitioner could even upload the irreducible loss of each point to a repository for others to use and improve on. This way, other practitioners need not train their own irreducible loss models."
> > > >
> > > > &nbsp;
> > > >
> > > > Have we addressed your remaining concerns with this clarification?

---

> > > > > ### Author Response · Authors · 2021-12-01
> > > > > **Changes to pdf**
> > > > >
> > > > > [Main changes in the PDF:]
> > > > > * Added Section 5.3 and Figure 5 on cheap irreducible loss models.
> > > > > * Added Clothing-1M in Figure 3 and Section 5, paragraph 1, and Section 5.2
> > > > > * Re _“Approximation from 4 to 6 might not be accurate if the hold-out set is comparable or smaller than the training data. If the hold-out set is large then why would one not use that for training the model in the first place?”_
> > > > >   * Addressed by Section 5.3
> > > > > * Re _“It would be interesting to add some discussion on how is the irreducible model is selected. Some ablation experiments showing the differences from equation 4 to equation 6 can be insightful.”_
> > > > >   * Ablation in Appendix D
> > > > >   * Selection of the irrloss model in Appendix C (experiment details)
> > > > > * Improved the precision and clarity of our theory section (Section 3)

---

### Author Response · Authors · 2021-11-25
**Thank you to reviewers**


Dear reviewers,

You provided excellent suggestions that have considerably improved the paper. Our response is extensive, with 4 new figures, a new large dataset, and a new section on training cheap irreducible loss models (with over 100 new experiment runs). We believe it addresses all concerns raised.

We would be grateful if you let us know if we have addressed your concerns, and if there are other concerns left to address.

Thank you for your time and best wishes,

The authors

 \
**Summary of changes:**



* **Costs of training irreducible loss model:** Reviewers NdhY and DHeZ had concerns on the costs of training the irreducible loss model. We added an extensive new section (5.3) reporting results of using cheap irreducible loss models–– models trained with minimal compute or data–– in a variety of scenarios.
* **Large-scale web-scraped data:** Reviewers Yofy, and DHeZ were interested to see the performance of our method on a larger dataset. In section 5.2, we report new results on Clothing-1M, a widely accepted benchmark with over 1 million web-scraped, noisy labels. While baselines were not robust on this dataset, our method provides a large speedup.
* **Additional new experiments:**
    * In response to reviewer BBTj, we have compared our method to a number of active learning baselines in appendix G, where our method outperforms all active learning baselines.
    * We ablate our approximation.
* Finally, we respond to feedback on clarity, novelty, and theory. We also added an ethics statement.

---

### Decision · Program_Chairs · 2022-01-20

**Decision:**

Reject

**Comment:**

The paper provides a method to accelerate training by choosing a subset of points. After the initial submission, the reviewers raised a major concern about the practicality of the method. In the rebuttal phase the authors provided additional experiments on a large datasets that addressed this concern. That being said, the reviews are still quite borderline. The biggest remaining concern is about the quality of writing. Specifically, there are still requests to “fix the narrative” (NdhY, DHeZ). In addition, some details seem to remain vague regarding the positioning of the paper when compared to the active learning literature (BBTj).
Overall, the paper seems to have potential, especially with the new experiments. However, the changes it required when compared to the originally submitted version are simply too extensive to be thoroughly reviewed in a rebuttal phase.